# Full-Endoscopic Lumbar Foraminotomy for Foraminal Stenosis in Spondylolisthesis: Two-Year Follow-Up Results

**DOI:** 10.3390/diagnostics12123152

**Published:** 2022-12-13

**Authors:** Do Yeon Rhee, Yong Ahn

**Affiliations:** Department of Neurosurgery, Gil Medical Center, Gachon University College of Medicine, Incheon 21565, Republic of Korea

**Keywords:** foraminotomy, full-endoscopic, lumbar, spondylolisthesis, stenosis

## Abstract

Full-endoscopic lumbar foraminotomy (FELF) under local anesthesia has been developed as a minimally invasive alternative for lumbar foraminal stenosis. Some authors have described this technique for treating various lumbar spondylolisthesis. However, few studies have reported the outcomes of FELF for foraminal stenosis in patients with stable spondylolisthesis. This study aimed to demonstrate the specific technique and clinical outcomes of FELF for foraminal stenosis in patients with spondylolisthesis. Twenty-three consecutive patients with foraminal stenosis and stable spondylolisthesis were treated with FELF. Among them, 21 patients were followed up for 2 years. Full-endoscopic foraminal decompression via the transforaminal approach was performed by a senior surgeon. Clinical outcomes were evaluated using the visual analog pain score (VAS), Oswestry Disability Index (ODI), and modified MacNab criteria. The VAS and ODI scores significantly improved at the two-year follow-up. The global effects were excellent in six patients (28.6%), good in 13 (61.9%), and fair in two (9.5%). Therefore, all patients showed clinical improvement, with a success (excellent/good) rate of 90.5%. No significant surgical complications or signs of further instability were observed. FELF can be used for foraminal stenosis in patients with stable spondylolisthesis. A specialized surgical technique is required for foraminal decompression of spondylolisthesis.

## 1. Introduction

Currently, lumbar decompression with fusion is the gold standard surgical option for intractable radiculopathy due to lumbar spondylolisthesis. However, extensive open surgery may cause significant morbidity or long-term sequelae, particularly in patients with underlying medical problems. Therefore, minimally invasive spine surgery (MISS) is required to reduce the risk of extensive fusion surgery. Some authors have recently reported that stable degenerative spondylolisthesis with stenosis can be treated with adequate decompression alone [1,2,3].

Since Kambin et al., Yeung et al., and Knight et al. independently reported percutaneous endoscopic lumbar foraminoplasty techniques using different methods [4,5,6], the full-endoscopic lumbar foraminotomy (FELF) technique has been developed for foraminal decompression under endoscopic visualization, via the transforaminal approach [7,8,9]. This novel foraminal decompression method has evolved using different surgical devices such as micropunches, bone reamers, side-firing lasers, and endoscopic burrs. Although the clinical results of current FELF are comparable to those of open foraminotomy owing to technical advancements, the learning curve is steep, and the entry barrier is still high for standard spine surgeons [9,10,11].

The FELF technique has also been applied to degenerative or isthmic lumbar spondylolisthesis [12,13,14,15,16,17,18,19]. Most trials have been performed for lumbar intracanal stenosis or herniated discs with spondylolisthesis. Few studies have demonstrated a technique for foraminal stenosis with stable spondylolisthesis. This article will help aspiring endoscopic surgeons understand the standard method of FELF and the added technical keys for foraminal decompression in spondylolisthesis. We aimed to demonstrate the clinical results of FELF for foraminal stenosis in stable spondylolisthesis and to discuss technical knowledge.

## 2. Materials and Methods

### 2.1. Patient Evaluation

Twenty-three consecutive patients with foraminal stenosis with stable spondylolisthesis were treated with FELF. Among them, 21 patients (91.3%) were surveyed in this two-year follow-up study. Patients were prospectively registered in our database, and their medical records were retrospectively assessed. The study was conducted in accordance with the Declaration of Helsinki, and approved by the Institutional Review Board, and written informed consent was obtained from all patients.

We selected patients according to the strict inclusion criteria as follows: (1) unilateral, unbearable radicular leg pain despite more than 3 months of nonoperative therapies; (2) nerve root compression due to severe foraminal stenosis [20,21], with single-level spondylolisthesis demonstrated on computed tomography (CT) scans and magnetic resonance imaging (MRI); (3) stable lumbar spondylolisthesis without significant segmental hypermobility on dynamic lateral X-rays; and (4) foraminal stenosis, as the source of radiculopathy, which was verified by thorough neurologic examinations and repeated selective nerve root block to the exiting nerve root (ENR).

Patients with complaints of low back pain, segmental instability or hypermobility, acute herniated lumbar disc, severe central stenosis, and other pathological conditions, such as inflammation, infection, trauma, or tumor, were excluded from the study. Cases of suspicious or different level pathology were also excluded.

Clinical data were obtained through regular outpatient visits and telephonic interviews. Clinical results were evaluated using the visual analog pain score (VAS) and the Oswestry disability index (ODI). The global effects in the final review were assessed using the modified MacNab criteria.

### 2.2. Surgical Procedure

All procedures were performed under local anesthesia based on the standard FELF method [22,23] and additional techniques specific to spondylolisthesis. It consisted of three steps: (1) transforaminal approach under fluoroscopic view, (2) foraminal widening by bone resection, and (3) ENR release by soft tissue removal (Figure 1).

As a premedication, intramuscular midazolam (0.05 mg/kg) and intravenous fentanyl (0.8 μg/kg) were administered on call. The patient was kept at a conscious sedation status with monitoring of any changes in symptoms and signs during the procedure. The patient was prone, with their knees flexed on a radiolucent spine table.

#### 2.2.1. Transforaminal Approach

This step aimed to ensure the safe engagement of the working sheath at the foraminal zone. The outside-in approach is better than the inside-out technique because the foraminal safe working zone is narrow. The opening of the working sheath was located immediately in front of the ENR.

The skin entry point and access angle were determined using CT, MRI, and X-rays. The typical approach angle is recommended to be steep (45° or more) for foraminal decompression and can be adjusted according to the pathological point. An 18-gauged needle was inserted posterolaterally into the foraminal zone under fluoroscopic guidance. The needle tip was inserted deeply into the foraminal disc or on the vertebral body, touching the surface of the superior articular process (SAP). A guidewire replaced the needle, and an obturator proceeded until its head was fitted at the foramen without any signs of access pain or neural irritation. The bevel-ended working sheath was advanced along the obturator by gentle tapping with a mallet and placed firmly in the foraminal zone with its sharp end away from the ENR. The surgical field for foraminal decompression was then created with a working sheath outside the foramen. The ENR was seen in the endoscopic view without irritation during the entire procedure (Figure 2A).

#### 2.2.2. Endoscopic Bone Work

This step aimed to widen the foraminal dimension by resecting the offending bone structures after the insertion of a working channel rigid endoscope. The initial endoscopic visualization included ENR with perineural fat and disc surface (Figure 2B). These structures helped the surgeon maintain the correct spatial orientation during the procedure. The lateral surface of the SAP was exposed by rotating the working sheath and endoscope. The tip of the SAP was then drilled using various endoscopic burrs along the ENR until the ligamentum flavum and foraminal ligaments at the axillary zone were sufficiently exposed. Any bleeding from the bone or venous plexus was coagulated using radiofrequency tips and hemostatic agents. In advanced spondylolisthesis with collapsed disc space, the ENR can be pinched by a narrow space between the upper pedicle and lower vertebral endplates rather than by the SAP. Therefore, the ENR can be released by sculpturing these bony structures. Bone resection is an essential and critical process of foraminal decompression specific to spondylolisthesis cases (Figure 2C).

#### 2.2.3. Endoscopic Soft Tissue Work

This step aimed to release the ENR by removing the ligamentum flavum, foraminal ligaments, and, optionally, herniated discs. After sufficient bone resection, sophisticated soft tissue removal was performed, and the ENR was released. Decompression was directed proximally, and the nerve root course was traced until the axillary epidural zone was exposed. Hypertrophic ligaments and extruded disc fragments were removed using micropunches, forceps, and radiofrequency tips. Even minor bleeding can seriously interfere with the endoscopic surgical field. Therefore, meticulous hemostasis is essential to ensure a clear vision during the procedure. The ENR began appearing and was gradually released freely as soft tissue work proceeded. At this time, the surgeon should be careful not to damage the dural membrane. The tissue debris was cleared with radiofrequency, and the neural tissues were separated from the offending tissues. The axillary epidural zone may be a key landmark in foraminal decompression. Exposure of the dural sac and the starting point of ENR indicated successful foraminal decompression. Once the proximal axillary zone was confirmed and released, the nerve root was examined in the lateral exit zone. Any remaining ligament or disc tissue was trimmed during the full-scale foraminal decompression.

#### 2.2.4. Finishing Point

Finally, defining the correct finishing point is necessary to prevent incomplete decompression. The endpoint of FELF can be determined by sufficient exposure, free mobilization, and strong pulsation of neural tissues (Figure 2D). Postoperatively, the surgeon monitored the patient’s condition and pain status for at least 3 h. The patient could be discharged within 24 h in the absence of any adverse events (Figure 3 and Figure 4).

### 2.3. Statistical Analysis

Statistical analyses were conducted between the preoperative and postoperative clinical data using repeated-measures analysis of variance and a paired *t*-test. Statistical significance was set at *p* < 0.05.

## 3. Results

The mean age of the 21 patients (13 women and eight men) was 69.5 years (range: 53–83 years). The BMI (mean ± SD) was 22.91 ± 2.89 kg/m^2^. The operative levels were L5-S1 in 11 (52.4%) patients, L4-5 in eight (38.1%), and L3-4 in two (9.5%). The degree of spondylolisthesis was grade 1 in 19 patients (90.5%) and grade 2 in two (9.5%). The mean operative time was 61.4 min (range, 35–115 min). The mean postoperative hospital stay was 1.8 days (range, 1–5).

The preoperative VAS score for the lumbar radiculopathy (mean ± SD) was 7.95 ± 0.74, which improved to 2.62 ± 0.80, 1.91 ± 0.44, 1.48 ± 0.60, and 1.52 ± 0.60 at 6 weeks, 6 months, 1 year, and 2 years postoperatively, respectively (*p* < 0.001) (Figure 5A). The preoperative ODI (mean ± SD) was 75.14 ± 8.40%, which improved to 29.20 ± 6.24%, 22.75 ± 6.83%, 17.33 ± 6.79%, and 18.19 ± 7.45% at 6 weeks, 6 months, 1 year, and 2 years postoperatively, respectively (*p* < 0.001) (Figure 5B). The global results based on the modified Macnab criteria were as follows: excellent in six patients (28.6%), good in 13 (61.9%), and fair in two (9.5%). Therefore, all patients showed clinical improvement, with a success (excellent/good) rate of 90.5% (Figure 6).

Two patients experienced transient postoperative dysesthesia (POD). Symptoms improved with oral medication within 4 weeks, and no other significant perioperative complications were noted. Moreover, no further low back pain or radiological signs of progressive instability were noted during the follow-up period.

## 4. Discussions

### 4.1. Diagnosis and Clinical Outcomes

The primary benefit of FELF is its minimal invasiveness under local anesthesia. Therefore, this technique may be suitable for older adults or medically compromised patients. The mean age of the patients (69.5 years) was relatively higher than that of other series of endoscopic spine surgeries. Most patients have chronic radiculopathy due to a long history of spondylolisthesis. Moreover, they are frequently excluded from extensive fusion surgery under general anesthesia either by themselves or by surgeons. We confirmed that radiculopathy was involved in severe foraminal stenosis with spondylolisthesis through neurological examination, imaging, and preoperative selective nerve root block. We identified the source of the pain using at least three different methods. Regarding imaging studies, only ‘severe’ foraminal stenosis cases with spondylolisthesis on MRI [20,21] were indicated for the FELF technique. Subsequently, all patients underwent one or more selective nerve root blocks and were able to verify the source of radiculopathy.

Perioperative data revealed the minimal invasiveness of FELF. The mean operative time (61.4 min with negligible blood loss was less than that of fusion surgery [24,25,26,27]. The mean postoperative hospital stays of 1.8 days were also straightforward and could facilitate an earlier return to work. Postoperative clinical outcomes improved in terms of both pain scores and functional status. The mean reduction in the VAS score was 6.43 at the final review (*p* < 0.001), and the mean improvement in the ODI was 56.95 at the final evaluation (*p* < 0.001). An improvement of >50% in the VAS score [28] or 30% in the ODI [29,30] after a surgical procedure is regarded as clinically relevant. The global success rate (excellent or good) of 90.5% with symptomatic improvement in all patients was comparable to that of open foraminotomy [31,32,33,34,35,36,37]. None of the patients experienced any significant complications or signs of further instability during the follow-up period. Therefore, our data indicated that the effects of FELF on foraminal stenosis in stable spondylolisthesis are relevant and practical.

Given the inherent characteristics of endoscopic spine surgery, clinical results may depend on the surgeon’s skill and patient selection. Therefore, a systematic training course with a steep learning curve is required to improve technical proficiency. Once mastered, the endoscopic procedure can result in reliable and relevant outcomes.

### 4.2. Evolution of Full-Endoscopic Lumbar Foraminotomy

Current FELF has become a practical foraminal decompression procedure after long-term technical evolution. First-generation FELF was a laser foraminoplasty introduced by Knight et al. [5,38]. They used a side-firing laser to ablate foraminal ligaments and shoulder osteophytes under endoscopic visualization. The second-generation technique uses a bony reamer to sculpt the SAP. Ahn et al. [7] introduced percutaneous endoscopic foraminotomy using a bone trephine and Ho:YAG side-firing laser. Schubert and Hoogland [8] applied a foraminoplasty method using bone trephine to remove migrated disc fragments. Owing to the development of decompression devices, current or third-generation techniques can decompress the stenotic foramen as effectively as an open foraminotomy. Endoscopic spine surgeons can use a variety of valuable instruments, such as endoscopic punches, burrs, and steerable devices, in the working channel under direct endoscopic visualization [9,39].

### 4.3. Current Full-Endoscopic Foraminotomy for Spondylolisthesis and Its Benefits

Since Knight et al. introduced endoscopic laser foraminoplasty for isthmic spondylolisthesis [12], some pioneers have reported the effects of transforaminal endoscopic surgery for spondylolisthesis [13,14,15,16,17,18,19]. However, few studies have described the technical processes or tips for foraminal decompression in patients with stable spondylolisthesis. This study demonstrated the precise technical aspects of endoscopic foraminal decompression specific to advanced spondylolisthesis cases, as well as the surgical outcomes.

One of the most peculiar benefits of FELF is that it is feasible to perform effective foraminal decompression without additional instrumentation and fusion. Our data showed no further slippage or instability during follow-up reviews. This technique may be helpful in patients who refuse fusion surgery or in medically compromised older patients because the procedure can be performed percutaneously under local anesthesia. Eventually, surgical complications of extensive fusion surgery can be avoided, and the patient can return to their ordinary life earlier.

However, the steep learning curve and technical difficulties may be the critical disadvantages of this technique. Only after technical proficiency has been achieved can the endoscopic surgeon perform FELF. Therefore, the clinical use of FELF for spondylolisthesis should be carefully considered. Considering the inherent technical limitation, appropriate patient selection or clinical application of FELF is also essential for clinical success. We recommend conventional open surgery rather than endoscopic surgery for patients with concurrent severe central stenosis or high-grade slippage of grade 3 or more.

### 4.4. Technical Keys Specific to Foraminal Stenosis with Spondylolisthesis

The primary pathologies for typical lumbar foraminal stenosis are hypertrophic SAP and ligamentum flavum compressing the ENR. Therefore, the removal of the SAP and ligamentous structures to release ENR is the final process of the FELF technique. However, foraminal stenosis in patients with advanced spondylolisthesis differs significantly. Unlike typical foraminal stenosis, ENR may be pinched by a narrow space between the upper pedicle and lower vertebral endplates, as well as SAP. To decompress the pinched ENR, the surgeon should target the point between the disc and the lower vertebral endplate during the initial transforaminal approach. After placing the working sheath, the surgeon identifies the ENR compressed by the lower vertebral endplates and SAP, possibly the upper pedicle. Full-scale foraminal decompression could be achieved only after sculpturing those bony walls and releasing the axillary starting point of the ENR.

### 4.5. Limitations of the Study

This study had several inherent limitations. First, the analysis was performed retrospectively without an appropriate control group. Therefore, considerable selection bias may have existed in the enrollment criteria. Second, the number of patients was too small to draw a conclusive result. Finally, the two-year follow-up period may be short to assess whether further slippage or instability develops over time. Therefore, a long-term, prospective, randomized trial or comparative cohort study comparing FELF and open fusion surgery with a more significant number of cases is required.

## 5. Conclusions

FELF can be effective in well-selected cases of foraminal stenosis in stable lumbar spondylolisthesis. This would be helpful for elderly and medically compromised patients at risk for extensive fusion surgery. A specialized technique unique to spondylolisthesis is essential to obtain a relevant outcome.

## Figures and Tables

**Figure 1 diagnostics-12-03152-f001:**
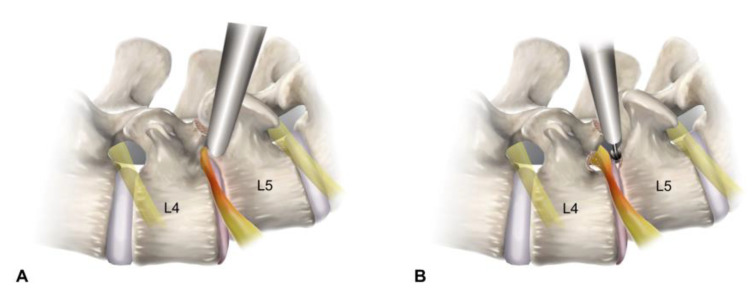
Schematic illustrations depicting the surgical procedure of full-endoscopic lumbar foraminotomy for foraminal stenosis with spondylolisthesis at the L4-L5 level. (**A**) Placement of the working sheath viewing the foraminal surgical field, avoiding the exiting nerve root (outside-in approach). (**B**) Bone works using endoscopic burrs for resecting the superior articular process and lower vertebral endplate. (**C**) Soft tissue decompression with removal of the ligamentum flavum and extruded disc. (**D**) End point of the full-scale foraminal decompression from the axillary to the lateral exit zone.

**Figure 2 diagnostics-12-03152-f002:**
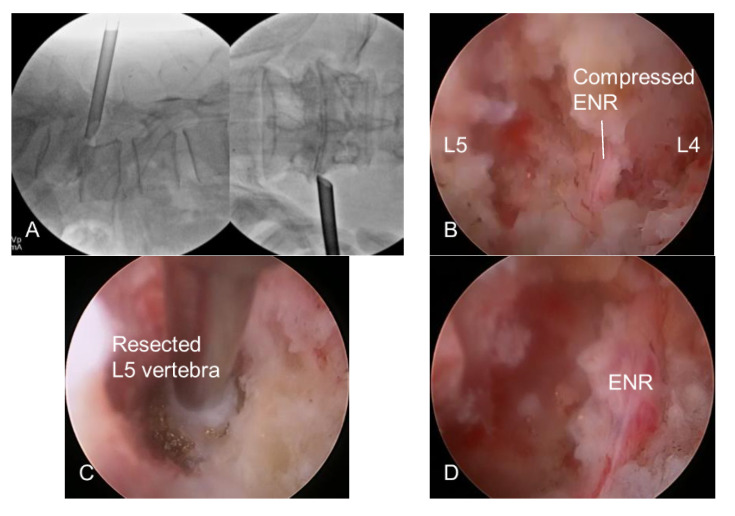
Intraoperative fluoroscopic and endoscopic views. (**A**) Foraminal docking of the working sheath. (**B**) Initial endoscopic view. Note the exiting nerve root (ENR) compressing by the surrounding bone and soft tissues. (**C**) Foraminal unroofing with the removal of bony structures including the lower vertebral endplate compressing the ENR. (**D**) At the final view, the ENR was freely released along the entire route of the nerve root.

**Figure 3 diagnostics-12-03152-f003:**
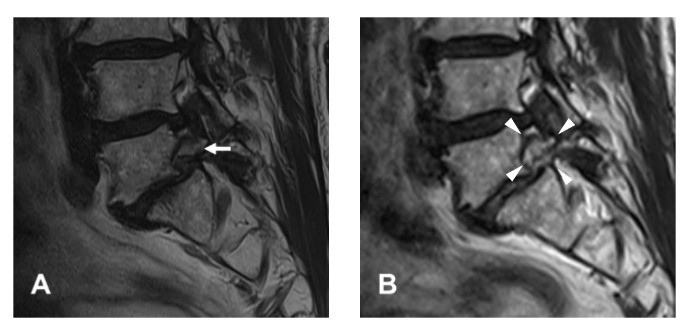
An illustrative case of a 65-year-old male patient. (**A**) Preoperative magnetic resonance image (MRI) showing foraminal stenosis with spondylolisthesis at the L5-S1 level (arrow). (**B**) Postoperative MRI showing foraminal decompression after resection of the bone and soft tissues (arrowheads).

**Figure 4 diagnostics-12-03152-f004:**
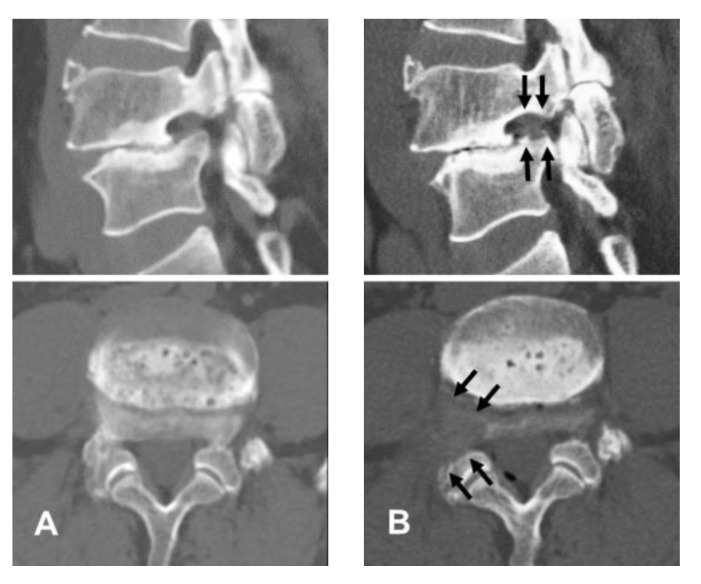
An illustrative case of a 61-year-old male patient. (**A**) Preoperative computed tomography (CT) images showing right-side foraminal stenosis in spondylolisthesis with collapsed disc space at the L4-5 level. (**B**) Postoperative CT images showing foraminal decompression by removal of bony structures, including the lower vertebral endplate and superior articular process (arrows).

**Figure 5 diagnostics-12-03152-f005:**
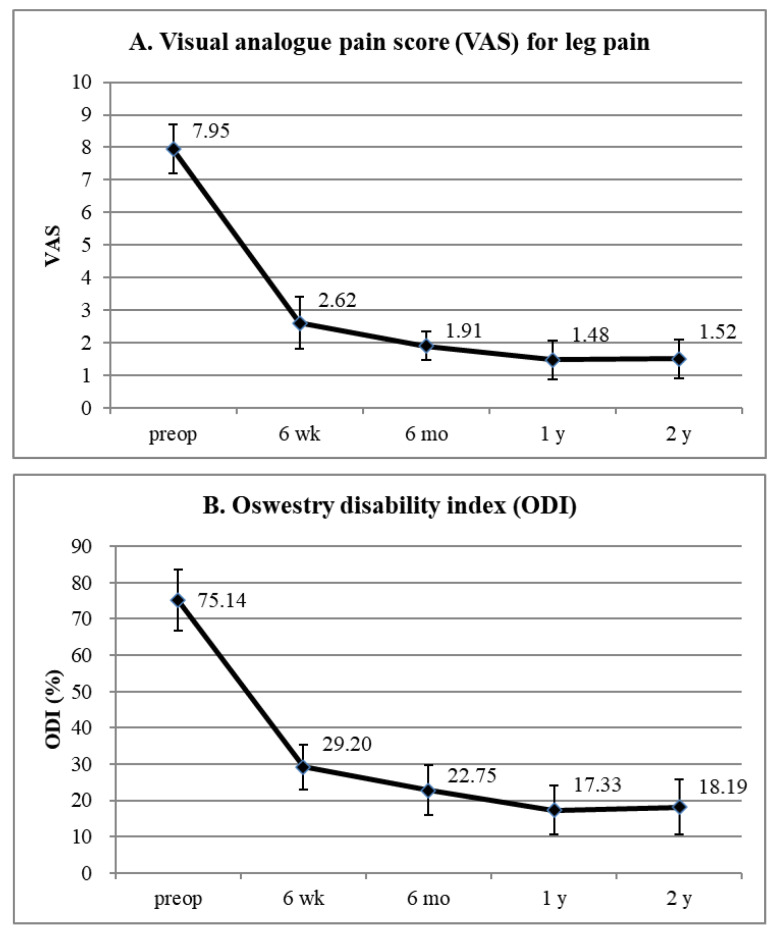
Clinical results. (**A**) Visual analog pain scale for radicular pain preoperatively and at 6 weeks, 6 months, 1 year, and 2 years postoperatively. (**B**) Oswestry disability index scores preoperatively and at 6 weeks, 6 months, 1 year, and 2 years postoperatively.

**Figure 6 diagnostics-12-03152-f006:**
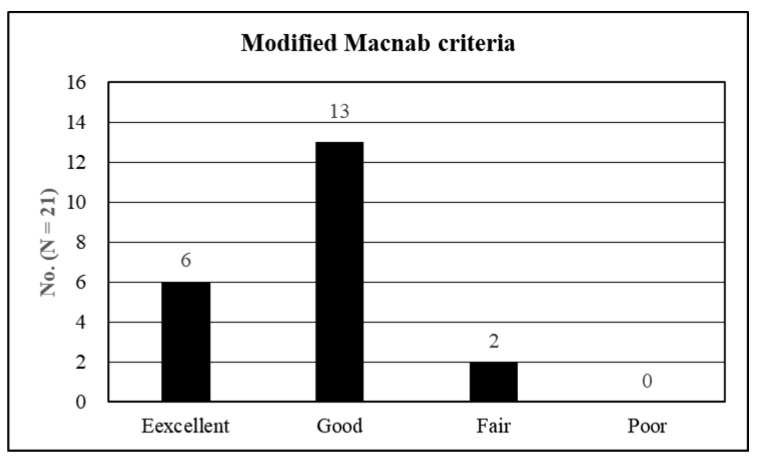
The global results based on the modified Macnab criteria were as follows: excellent in six patients (28.6%), good in 13 (61.9%), and fair in two (9.5%). Therefore, all patients showed clinical improvements, and the successful (excellent/good) rate was 90.5%.

## Data Availability

The data presented in this study are available on request from the corresponding author.

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
