# Peer review of "Full-Endoscopic Lumbar Foraminotomy for Foraminal Stenosis in Spondylolisthesis: Two-Year Follow-Up Results"

_diagnostics, 2022, doi:10.3390/diagnostics12123152_

Round 1

Reviewer 1 Report

I believe that good clinical results may depend on the surgeon's skill and patient selection. The FELF technique is a good and reasonable idea for stable spondylolisthesis. For this technique. I hope the author could mention more detail about the part of resection of the upper pedicle because it is a relatively risky procedure, especially under local anesthesia.

From the author's previous study "Ahn, Y.; Lee, S.G. Percutaneous endoscopic lumbar foraminotomy: how I do it. Acta. Neurochir. (Wien) 2022, 164, 933-936." the article didn't mention about the resection of the upper pedicle. 

Is it necessary to do this procedure (removal of the upper pedicle )for every case? how much bony part should be resected?  How many percent of patients received the procedure? I hope the author could give the readers some suggestions:  when we should do the resection of the upper pedicle and when we should just do the soft tissue work and lower vertebral endplate.

And I hope the author could provide the pre-op and post-op images (CT or MRI) that could show the bony procedure more clearly.

Author Response

Response to Reviewers

Reviewer 1

I believe that good clinical results may depend on the surgeon's skill and patient selection. The FELF technique is a good and reasonable idea for stable spondylolisthesis. For this technique. I hope the author could mention more detail about the part of resection of the upper pedicle because it is a relatively risky procedure, especially under local anesthesia.

From the author's previous study "Ahn, Y.; Lee, S.G. Percutaneous endoscopic lumbar foraminotomy: how I do it. Acta. Neurochir. (Wien) 2022, 164, 933-936." the article didn't mention about the resection of the upper pedicle. 

Is it necessary to do this procedure (removal of the upper pedicle )for every case? how much bony part should be resected?  How many percent of patients received the procedure? I hope the author could give the readers some suggestions:  when we should do the resection of the upper pedicle and when we should just do the soft tissue work and lower vertebral endplate.

Response 1:

Thank you for your constructive comments. Removal of the upper pedicle was only necessary for some cases. In the case of advanced spondylolisthesis with collapsed disc space, for example, we had to remove a part of the upper pedicle to release the ENR at the proximal axillary zone and to decompress the lower vertebral endplate at the distal part of the ENR for the full-scale foraminal decompression. This process is not necessary for FELF for usual foraminal stenosis. We revised the surgical procedure part and discussion section to clarify this point.

2.2.2. Endoscopic bone work

“In advanced spondylolisthesis with collapsed disc space, the ENR can be pinched by a narrow space between the upper pedicle and lower vertebral endplates rather than by the SAP.” (Lines 120-122)

4.4. Technical keys specific to foraminal stenosis with spondylolisthesis

Unlike typical foraminal stenosis, ENR may be pinched by a narrow space between the upper pedicle and lower vertebral endplates, as well as SAP. (Lines 276-278) To decompress the pinched ENR, the surgeon should target the point between the disc and the lower vertebral endplate during the initial transforaminal approach. After placing the working sheath, the surgeon identifies the ENR compressed by the lower vertebral endplates and SAP, possibly the upper pedicle. Full-scale foraminal decompression could be achieved only after sculpturing those bony walls and releasing the axillary starting point of the ENR. “(Lines 280-283)

And I hope the author could provide the pre-op and post-op images (CT or MRI) that could show the bony procedure more clearly.

Response 2:

According to your recommendation, we added another illustrated case with pre- and postoperative CT scans.

“Figure 4. An illustrative case of a 61-year-old male patient. (A) Preoperative computed tomography (CT) images showing right-side foraminal stenosis in spondylolisthesis with collapsed disc space at the L4-5 level. (B) Postoperative CT images showing foraminal decompression by removal of bony structures, including the lower vertebral endplate and superior articular process (arrows).” (Lines 165-168)

Reviewer 2 Report

This is a very interesting study about the growing field of endoscopy in Spine surgery

However I may have some concern about the manuscript 

Patient Selection and Evaluation: 

All patients had selective nerve block to assess the level of compression and the source of radiculopathy. 

Nothing is said about patients who may have bilateral symptoms, as non-rarely seen in spondylolisthesis: what about them?

Similarly Spondylolisthesis, especially when degenerative, can be a multi level disease: Were there any patient operated by this technique at different level? 

Finally, back pain common symptom usually associated with radiculopathy in spondylolisthesis: Has it been evaluated by a simple VAS? It could also be a limitation compared to fusion 

Surgical procedure   

The authors described their efficient transforaminal outside-in approach. They mentioned that in advanced spondylolisthesis, the ENR could be narrowed by the upper pedicle. In that case, in the microsurgical approach, to release the root shoulder, it can be useful to decompress the upper adjacent level.

Did the author meet that situation? Have they observed any insufficient decompression by their approach? It should be mention in the discussion. 

Overall, limitations of the surgical procedure, over the learning curve, should be discussed

Local anesthesia 

The authors argue that every patient has been operated under local anesthesia, which is undoubtedly an major advantage

However, it is written that " Intramuscular midazolam and fentanyl were administered intravenously" This is unclear. If there is any IV drug, patient monitoring is necessary like in general anesthesia. This should be clarified 

Many of such patients are also operated under spinal anesthesia with excellent outcomes. 

Author Response

Reviewer 2

This is a very interesting study about the growing field of endoscopy in Spine surgery

However I may have some concern about the manuscript 

Patient Selection and Evaluation: 

All patients had selective nerve block to assess the level of compression and the source of radiculopathy. 

Nothing is said about patients who may have bilateral symptoms, as non-rarely seen in spondylolisthesis: what about them?

Similarly Spondylolisthesis, especially when degenerative, can be a multi level disease: Were there any patient operated by this technique at different level? 

Finally, back pain common symptom usually associated with radiculopathy in spondylolisthesis: Has it been evaluated by a simple VAS? It could also be a limitation compared to fusion.

Response 1:

Thank you for your constructive comments. We agree with your points. We selected patients according to the strict inclusion criteria. Patients with single-level spondylolisthesis who suffered unilateral radicular leg pain without significant back pain were selected for the FELF technique. Patients with spondylolisthesis might have complained of bilateral symptoms and back pain in their past history. However, many patients with chronic stable spondylolisthesis could have only radicular pain after a long-standing disease process. Patients with segmental instability or back pain should have decompression and fusion surgery.

The pain source was verified via multimodal diagnostic methods: 1) thorough neurological examination, 2) imaging studies including MRI, CT scans, and dynamic X-rays, and 3) repeated selective nerve root blocks. Only after the exact pain source had been verified was the procedure performed. Patients with bilateral symptoms and concurrent back pain were excluded. Cases of suspicious or different-level pathology were also excluded.

We revised the patient selection part as follows:

“We selected the patients according to the strict inclusion criteria as follows: 1) unilateral, unbearable radicular leg pain despite more than 3 months of nonoperative therapies; 2) nerve root compression due to severe foraminal stenosis [20,21], with single-level spondylolisthesis demonstrated on computed tomography (CT) scans and magnetic resonance imaging (MRI); 3) stable lumbar spondylolisthesis without significant segmental hypermobility on dynamic lateral X-rays; and 4) foraminal stenosis, as the source of radiculopathy, which was verified by thorough neurologic examinations and repeated selective nerve root block to the exiting nerve root (ENR).

Patients with complaints of low back pain, segmental instability, or hypermobility, acute herniated lumbar disc, severe central stenosis, and other pathological conditions, such as inflammation, infection, trauma, or tumor, were excluded from the study. Cases of suspicious or different level pathology were also excluded.” (Lines 59-70)

Surgical procedure   

The authors described their efficient transforaminal outside-in approach. They mentioned that in advanced spondylolisthesis, the ENR could be narrowed by the upper pedicle. In that case, in the microsurgical approach, to release the root shoulder, it can be useful to decompress the upper adjacent level.

Did the author meet that situation? Have they observed any insufficient decompression by their approach? It should be mention in the discussion. 

Overall, limitations of the surgical procedure, over the learning curve, should be discussed

 Response 2:

We found that the upper pedicle could compress the ENR in some advanced spondylolisthesis with collapsed disc space. In those cases, we had to remove some parts of the upper pedicle. However, for low-grade spondylolisthesis, this process was not needed. The primary structures are still the SAP and the shoulder osteophyte of the lower vertebral body. Therefore, we revised the surgical procedure section as follows.

“2.2.2. Endoscopic bone work

“In advanced spondylolisthesis with collapsed disc space, the ENR can be pinched by a narrow space between the upper pedicle and lower vertebral endplates rather than by the SAP.” (Lines 120-122)

4.4. Technical keys specific to foraminal stenosis with spondylolisthesis

“Unlike typical foraminal stenosis, ENR may be pinched by a narrow space between the upper pedicle and lower vertebral endplates, as well as SAP. (Lines 276-278) To decompress the pinched ENR, the surgeon should target the point between the disc and the lower vertebral endplate during the initial transforaminal approach. After placing the working sheath, the surgeon identifies the ENR compressed by the lower vertebral endplates and SAP, possibly the upper pedicle. Full-scale foraminal decompression could be achieved only after sculpturing those bony walls and releasing the axillary starting point of the ENR. “(Lines 280-283)

We always confirmed the release of the starting point of the ENR from the dural sac in all cases. So, there was no incomplete decompression at the proximal part of the ENR.

According to your precious recommendation, we added the limitations of FELF over the learning curve in the discussion section as follows.

4.3. Current full-endoscopic foraminotomy for spondylolisthesis and its benefits

“Considering the inherent technical limitation, appropriate patient selection or clinical application of FELF is also essential for clinical success. We recommend conventional open surgery rather than endoscopic surgery for patients with concurrent severe central stenosis or high-grade slippage of grade 3 or more.” (Lines 268-271)

Local anesthesia 

The authors argue that every patient has been operated under local anesthesia, which is undoubtedly an major advantage

However, it is written that " Intramuscular midazolam and fentanyl were administered intravenously" This is unclear. If there is any IV drug, patient monitoring is necessary like in general anesthesia. This should be clarified 

Many of such patients are also operated under spinal anesthesia with excellent outcomes.

Response 3:

Thank you for your essential point. The primary dose of the premedication does not affect the patient’s respiration and general condition. The routine monitoring of the patient’s vital signs was maintained during the procedure, like other percutaneous procedures. Of course, FELF can be performed under spinal or general anesthesia as required. However, all patients in our series had no significant procedure pain or discomfort during the procedure under local anesthesia. Furthermore, the local anesthesia might benefit the patient’s feedback if the ENR was irritated by the approaching devices during the transforaminal approach.

We revised the surgical procedure section to increase the reader’s understanding:

“All procedures were performed under local anesthesia (Line 76) based on the standard FELF method [22,23] and additional techniques specific to spondylolisthesis. It consisted of three steps: 1) transforaminal approach under fluoroscopic view, 2) foraminal widening by bone resection, and 3) ENR release by soft tissue removal (Figure 1).

As a premedication, intramuscular midazolam (0.05 mg/kg) and intravenous fentanyl (0.8 μg/kg) were administered on call. The patient was kept at conscious sedation status with monitoring of any changes in symptoms and signs during the procedure. The patient was prone, with their knees flexed on a radiolucent spine table.” (Lines 80-83)

Editor’s recommendation

Response 1:

We also revised the legend of Figure 5 according to the editor’s recommendation.

Figure 5. Clinical results. (A) Visual analog pain scale for radicular pain preoperatively and at 6 weeks, 6 months, 1 year, and 2 years postoperatively. (B) Oswestry disability index scores preoperatively and at 6 weeks, 6 months, 1 year, and 2 years postoperatively.” (Lines 194-196)

Round 2

Reviewer 1 Report

It loos fine in its present form.